# Multiview Neural Surface Reconstruction by Disentangling Geometry and Appearance

**Lior Yariv**        **Yoni Kasten**        **Dror Moran**

**Meirav Galun**        **Matan Atzmon**        **Ronen Basri**        **Yaron Lipman**

Weizmann Institute of Science

`{lior.yariv, yoni.kasten, dror.moran, meirav.galun, matan.atzmon, ronen.basri, yaron.lipman}@weizmann.ac.il`

## Abstract

In this work we address the challenging problem of multiview 3D surface reconstruction. We introduce a neural network architecture that simultaneously learns the unknown geometry, camera parameters, and a neural renderer that approximates the light reflected from the surface towards the camera. The geometry is represented as a zero level-set of a neural network, while the neural renderer, derived from the rendering equation, is capable of (implicitly) modeling a wide set of lighting conditions and materials. We trained our network on real world 2D images of objects with different material properties, lighting conditions, and noisy camera initializations from the DTU MVS dataset. We found our model to produce state of the art 3D surface reconstructions with high fidelity, resolution and detail.

## 1 Introduction

Learning 3D shapes from 2D images is a fundamental computer vision problem. A recent successful neural network approach to solving this problem involves the use of a (neural) differentiable rendering system along with a choice of (neural) 3D geometry representation. Differential rendering systems are mostly based on ray casting/tracing [41, 33, 24, 26, 38, 27], or rasterization [28, 20, 10, 25, 4], while popular models to represent 3D geometry include point clouds [49], triangle meshes [4], implicit representations defined over volumetric grids [17], and recently also neural implicit representations, namely, zero level sets of neural networks [26, 33].

The main advantage of implicit neural representations is their flexibility in representing surfaces with arbitrary shapes and topologies, as well as being mesh-free (i.e., no fixed a-priori discretization such as a volumetric grid or a triangular mesh). Thus far, differentiable rendering systems with implicit neural representations [26, 27, 33] did not incorporate lighting and reflectance properties required for producing faithful appearance of 3D geometry in images, nor did they deal with trainable camera locations and orientations.

The goal of this paper is to devise an end-to-end neural architecture system that can learn 3D geometries from masked 2D images and rough camera estimates, and requires no additional supervision, see Figure 1. Towards that end we represent the color of a pixel as a differentiable function in the three unknowns of a scene: the geometry, its appearance, and the cameras. Here, appearance means collectively all the factors that define the surface light field, *excluding* the geometry, i.e., the surface bidirectional reflectance distribution function (BRDF) and the scene's lighting conditions. We call this architecture the Implicit Differentiable Renderer (IDR). We show that IDR is able to approximate the light reflected from a 3D shape represented as the zero level set of a neural network. The approach can handle surface appearances from a certain restricted family, namely, all surface light fields that can be represented as continuous functions of the point on the surface, its normal, and the viewing direction. Furthermore, incorporating a global shape feature vector into IDR increases its ability to handle more complex appearances (e.g., indirect lighting effects).

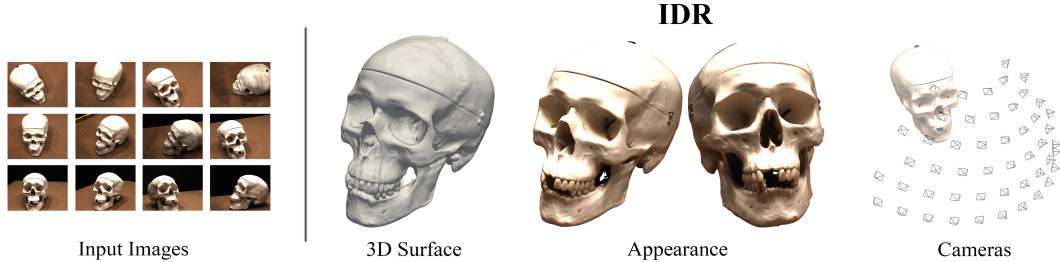

**IDR**

Input Images        3D Surface        Appearance        Cameras

Figure 1: We introduce IDR: end-to-end learning of geometry, appearance and cameras from images.

Most related to our paper is [33], that was first to introduce a fully differentiable renderer for implicit neural occupancy functions [31], which is a particular instance of implicit neural representation as defined above. Although their model can represent arbitrary color and texture, it cannot handle general appearance models, nor can it handle unknown, noisy camera locations. For example, we show that the model in [33], as-well-as several other baselines, fail to generate the Phong reflection model [8]. Moreover, we show experimentally that IDR produces more accurate 3D reconstructions of shapes from 2D images along with accurate camera parameters. Notably, while the baseline often produces shape artifact in specular scenes, IDR is robust to such lighting effects. Our code and data are available at `https://github.com/lioryariv/idr`.

To summarize, the key contributions of our approach are:

- End-to-end architecture that handles unknown geometry, appearance, and cameras.
- Expressing the dependence of a neural implicit surface on camera parameters.
- Producing state of the art 3D surface reconstructions of different objects with a wide range of appearances, from real-life 2D images, with both exact and noisy camera information.

## 2 Previous work

Differentiable rendering systems for learning geometry comes (mostly) in two flavors: differentiable rasterization [28, 20, 10, 25, 4], and differentiable ray casting. Since the current work falls into the second category we first concentrate on that branch of works. Then, we will describe related works for multi-view surface reconstruction and neural view synthesis.

**Implicit surface differentiable ray casting.** Differentiable ray casting is mostly used with *implicit* shape representations such as implicit function defined over a volumetric grid or implicit neural representation, where the implicit function can be the occupancy function [31, 5], signed distance function (SDF) [35] or any other signed implicit [2]. In a related work, [17] use a volumetric grid to represent an SDF and implement a ray casting differentiable renderer. They approximate the SDF value and the surface normals in each volumetric cell. [27] use sphere tracing of pre-trained DeepSDF model [35] and approximates the depth gradients w.r.t. the latent code of the DeepSDF network by differentiating the individual steps of the sphere tracing algorithm; [26] use field probing to facilitate differentiable ray casting. In contrast to these works, IDR utilize exact and differentiable surface point and normal of the implicit surface, and considers a more general appearance model, as well as handle noisy cameras.

**Multi-view surface reconstruction.** During the capturing process of an image, the depth information is lost. Assuming known cameras, classic Multi-View Stereo (MVS) methods [9, 40, 3, 45] try to reproduce the depth information by matching features points across views. However, a post-processing steps of depth fusion [6, 30] followed by the Poisson Surface Reconstruction algorithm [21] are required for producing a valid 3D watertight surface reconstruction. Recent methods use a collection of scenes to train a deep neural models for either sub-tasks of the MVS pipeline, e.g., feature matching [23], or depth fusion [7, 36], or for an End-to-End MVS pipeline [13, 47, 48]. When the camera parameters are unavailable, and given a set of images from a specific scene, Structure From Motion (SFM) methods [42, 39, 19, 16] are applied for reproducing the cameras and a sparse 3D reconstruction. Tang and Tan [44] use a deep neural architecture with an integrated differentiable Bundle Adjustment [46] layer to extract a linear basis for the depth of a reference frame, and features from nearby images and to optimize for the depth and the camera parameters in each forward pass. In contrast to these works, IDR is trained with images from a single target scene, producing an accurate watertight 3D surface reconstruction.

**Neural representation for view synthesis.** Recent works trained neural networks to predict novel views and some geometric representation of 3D scenes or objects, from a limited set of images with known cameras. [41] encode the scene geometry using an LSTM to simulate the ray marching process. [32] use a neural network to predict volume density and view dependent emitted radiance to synthesis new views from a set of images with known cameras. [34] use a neural network to learns the surface light fields from an input image and geometry and predicting unknown views and/or scene lighting. Differently from IDR, these methods do not produce a 3D surface reconstruction of the scene's geometry nor handle unknown cameras.

# 3   Method

Our goal is to reconstruct the geometry of an object from masked 2D images with possibly rough or noisy camera information. We have three unknowns: (i) *geometry*, represented by parameters $\theta \in \mathbb{R}^m$; (ii) *appearance*, represented by $\gamma \in \mathbb{R}^n$; and (iii) *cameras* represented by $\tau \in \mathbb{R}^k$. Notations and setup are depicted in Figure 2.

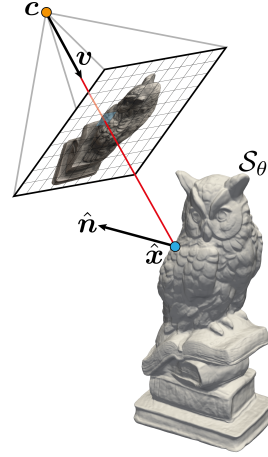

We represent the geometry as the zero level set of a neural network (MLP) $f$,

$$\mathcal{S}_\theta = \left\{ \boldsymbol{x} \in \mathbb{R}^3 \mid f(\boldsymbol{x}; \theta) = 0 \right\}, \tag{1}$$

with learnable parameters $\theta \in \mathbb{R}^m$. To avoid the everywhere 0 solution, $f$ is usually regularized [31, 5]. We opt for $f$ to model a signed distance function (SDF) to its zero level set $\mathcal{S}_\theta$ [35]. We enforce the SDF constraint using the implicit geometric regularization (IGR) [11], detailed later. SDF has two benefits in our context: First, it allows an efficient ray casting with the sphere tracing algorithm [12, 17]; and second, IGR enjoys implicit regularization favoring smooth and realistic surfaces.

**IDR forward model.** Given a pixel, indexed by $p$, associated with some input image, let $R_p(\tau) = \{\boldsymbol{c}_p + t\boldsymbol{v}_p \mid t \geq 0\}$ denote the ray through

Figure 2: Notations.

pixel $p$, where $\boldsymbol{c}_p = \boldsymbol{c}_p(\tau)$ denotes the unknown center of the respective camera and $\boldsymbol{v}_p = \boldsymbol{v}_p(\tau)$ the direction of the ray (i.e., the vector pointing from $\boldsymbol{c}_p$ towards pixel $p$). Let $\hat{\boldsymbol{x}}_p = \hat{\boldsymbol{x}}_p(\theta, \tau)$ denote the first intersection of the ray $R_p$ and the surface $\mathcal{S}_\theta$. The incoming radiance along $R_p$, which determines the rendered color of the pixel $L_p = L_p(\theta, \gamma, \tau)$, is a function of the surface properties at $\hat{\boldsymbol{x}}_p$, the incoming radiance at $\hat{\boldsymbol{x}}_p$, and the viewing direction $\boldsymbol{v}_p$. In turn, we make the assumptions that the surface property and incoming radiance are functions of the surface point $\hat{\boldsymbol{x}}_p$, and its corresponding surface normal $\hat{\boldsymbol{n}}_p = \hat{\boldsymbol{n}}_p(\theta)$, the viewing direction $\boldsymbol{v}_p$, and a global geometry feature vector $\hat{\boldsymbol{z}}_p = \hat{\boldsymbol{z}}_p(\hat{\boldsymbol{x}}_p; \theta)$. The IDR forward model is therefore:

$$L_p(\theta, \gamma, \tau) = M(\hat{\boldsymbol{x}}_p, \hat{\boldsymbol{n}}_p, \hat{\boldsymbol{z}}_p, \boldsymbol{v}_p; \gamma), \tag{2}$$

where $M$ is a second neural network (MLP). We utilize $L_p$ in a loss comparing $L_p$ and the pixel input color $I_p$ to simultaneously train the model's parameters $\theta, \gamma, \tau$. We next provide more details on the different components of the model in equation 2.

## 3.1   Differentiable intersection of viewing direction and geometry

Henceforth (up until section 3.4), we assume a fixed pixel $p$, and remove the subscript $p$ notation to simplify notation. The first step is to represent the intersection point $\hat{\boldsymbol{x}}(\theta, \tau)$ as a neural network with parameters $\theta, \tau$. This can be done with a slight modification to the geometry network $f$.

Let $\hat{\boldsymbol{x}}(\theta, \tau) = \boldsymbol{c} + t(\theta, \boldsymbol{c}, \boldsymbol{v})\boldsymbol{v}$ denote the intersection point. As we are aiming to use $\hat{\boldsymbol{x}}$ in a gradient descent-like algorithm, all we need to make sure is that our derivations are correct in value and first derivatives at the current parameters, denoted by $\theta_0, \tau_0$; accordingly we denote $\boldsymbol{c}_0 = \boldsymbol{c}(\tau_0)$, $\boldsymbol{v}_0 = \boldsymbol{v}(\tau_0)$, $t_0 = t(\theta_0, \boldsymbol{c}_0, \boldsymbol{v}_0)$, and $\boldsymbol{x}_0 = \hat{\boldsymbol{x}}(\theta_0, \tau_0) = \boldsymbol{c}_0 + t_0\boldsymbol{v}_0$.

**Lemma 1.** *Let $\mathcal{S}_\theta$ be defined as in equation 1. The intersection of the ray $R(\tau)$ and the surface $\mathcal{S}_\theta$ can be represented by the formula*

$$\hat{\boldsymbol{x}}(\theta, \tau) = \boldsymbol{c} + t_0\boldsymbol{v} - \frac{\boldsymbol{v}}{\nabla_{\boldsymbol{x}} f(\boldsymbol{x}_0; \theta_0) \cdot \boldsymbol{v}_0} f(\boldsymbol{c} + t_0\boldsymbol{v}; \theta), \tag{3}$$

*and is exact in value and first derivatives of $\theta$ and $\tau$ at $\theta = \theta_0$ and $\tau = \tau_0$.*

To prove this functional dependency of $\hat{x}$ on its parameters, we use implicit differentiation [1, 33], that is, differentiate the equation $f(\hat{x}; \theta) \equiv 0$ w.r.t. $v, c, \theta$ and solve for the derivatives of $t$. Then, it can be checked that the formula in equation 3 possess the correct derivatives. More details are in the supplementary. We implement equation 3 as a neural network, namely, we add two linear layers (with parameters $c, v$): one before and one after the MLP $f$. Equation 3 unifies the sample network formula in [1] and the differentiable depth in [33] and generalizes them to account for unknown cameras. The normal vector to $\mathcal{S}_\theta$ at $\hat{x}$ can be computed by:

$$\hat{n}(\theta, \tau) = \nabla_x f(\hat{x}(\theta, \tau), \theta) / \left\| \nabla_x f(\hat{x}(\theta, \tau), \theta) \right\|_2. \tag{4}$$

Note that for SDF the denominator is 1, so can be omitted.

### 3.2 Approximation of the surface light field

The surface light field radiance $L$ is the amount of light reflected from $\mathcal{S}_\theta$ at $\hat{x}$ in direction $-v$ reaching $c$. It is determined by two functions: The bidirectional reflectance distribution function (BRDF) describing the reflectance and color properties of the surface, and the light emitted in the scene (i.e., light sources).

The BRDF function $B(x, n, w^o, w^i)$ describes the proportion of reflected radiance (i.e., flux of light) at some wave-length (i.e., color) leaving the surface point $x$ with normal $n$ at direction $w^o$ with respect to the incoming radiance from direction $w^i$. We let the BRDF depend also on the normal $n$ to the surface at a point. The light sources in the scene are described by a function $L^e(x, w^o)$ measuring the emitted radiance of light at some wave-length at point $x$ in direction $w^o$. The amount of light reaching $c$ in direction $v$ equals the amount of light reflected from $\hat{x}$ in direction $w^o = -v$ and is described by the so-called rendering equation [18, 14]:

$$L(\hat{x}, w^o) = L^e(\hat{x}, w^o) + \int_\Omega B(\hat{x}, \hat{n}, w^i, w^o) L^i(\hat{x}, w^i)(\hat{n} \cdot w^i) \, dw^i = M_0(\hat{x}, \hat{n}, v), \quad (5)$$

where $L^i(\hat{x}, w^i)$ encodes the incoming radiance at $\hat{x}$ in direction $w^i$, and the term $\hat{n} \cdot w^i$ compensates for the fact that the light does not hit the surface orthogonally; $\Omega$ is the half sphere centered at $\hat{n}$. The function $M_0$ represents the surface light field as a function of the local surface geometry $\hat{x}, \hat{n}$, and the viewing direction $v$. This rendering equation holds for every light wave-length; as described later we will use it for the red, green and blue (RGB) wave-lengths.

We restrict our attention to light fields that can be represented by a continuous function $M_0$. We denote the collection of such continuous functions by $\mathcal{P} = \{M_0\}$ (see supplementary material for more discussion on $\mathcal{P}$). Replacing $M_0$ with a (sufficiently large) MLP approximation $M$ (neural renderer) provides the light field approximation:

$$L(\theta, \gamma, \tau) = M(\hat{x}, \hat{n}, v; \gamma). \tag{6}$$

Disentanglement of geometry and appearance requires the learnable $M$ to approximate $M_0$ *for all* inputs $x, n, v$ rather than memorizing the radiance values for a particular geometry. Given an arbitrary choice of light field function $M_o \in \mathcal{P}$ there exists a choice of weights $\gamma = \gamma_0$ so that $M$ approximates $M_0$ for all $x, n, v$ (in some bounded set). This can be proved using a standard universality theorem for MLPs (details in the supplementary). However, the fact that $M$ *can* learn the correct light field function $M_0$ does not mean it is *guaranteed* to learn it during optimization. Nevertheless, being able to approximate $M_0$ for arbitrary $x, n, v$ is a *necessary condition* for disentanglement of geometry (represented with $f$) and appearance (represented with $M$). We name this necessary condition *$\mathcal{P}$-universality*.

**Necessity of viewing direction and normal.** For $M$ to be able to represent the correct light reflected from a surface point $x$, i.e., be $\mathcal{P}$-universal, it has to receive as arguments also $v, n$. The viewing direction $v$ is necessary even if we expect $M$ to work for a fixed geometry; e.g., for modeling specularity. The normal $n$, on the other hand, can be memorized by $M$ as a function of $x$. However, for disentanglement of geometry, i.e., allowing $M$ to learn appearance independently from the geometry, incorporating the normal direction is also necessary. This can be seen in Figure 3: A renderer $M$ without normal information will produce the same light estimation in cases (a)

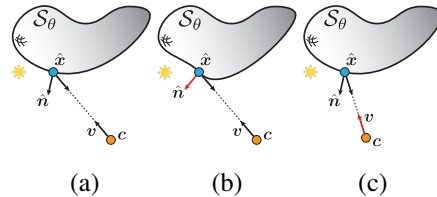

(a)      (b)      (c)

Figure 3: Neural renderers without $n$ and/or $v$ are not universal.

and (b), while a renderer $M$ without viewing direction will produce the same light estimation in cases (a) and (c). In the supplementary we provide details on how these renderers fail to generate correct radiance under the Phong reflection model [8]. Previous works, e.g., [33], have considered rendering functions of implicit neural representations of the form $L(\theta, \gamma) = M(\hat{x}; \gamma)$. As indicated above, omitting $n$ and/or $v$ from $M$ will result in a non-$\mathcal{P}$-universal renderer. In the experimental section we demonstrate that incorporating $n$ in the renderer $M$ indeed leads to a successful disentanglement of geometry and appearance, while omitting it impairs disentanglement.

**Accounting for global light effects.** $\mathcal{P}$-universality is a necessary conditions to learn a neural renderer $M$ that can simulate appearance from the collection $\mathcal{P}$. However, $\mathcal{P}$ does not include global lighting effects such as secondary lighting and self-shadows. We further increase the expressive power of IDR by introducing a global feature vector $\hat{z}$. This feature vector allows the renderer to reason *globally* about the geometry $\mathcal{S}_\theta$. To produce the vector $\hat{z}$ we extend the network $f$ as follows: $F(\boldsymbol{x}; \theta) = [f(\boldsymbol{x}; \theta), \boldsymbol{z}(\boldsymbol{x}; \theta)] \in \mathbb{R} \times \mathbb{R}^\ell$. In general, $\boldsymbol{z}$ can encode the geometry $\mathcal{S}_\theta$ relative to the surface sample $\boldsymbol{x}$; $\boldsymbol{z}$ is fed into the renderer as $\hat{\boldsymbol{z}}(\theta, \tau) = \boldsymbol{z}(\hat{\boldsymbol{x}}; \theta)$ to take into account the surface sample $\hat{\boldsymbol{x}}$ relevant for the current pixel of interest $p$. We have now completed the description of the IDR model, given in equation 2.

### 3.3 Masked rendering

Another useful type of 2D supervision for reconstructing 3D geometry are masks; masks are binary images indicating, for each pixel $p$, if the object of interest occupies this pixel. Masks can be provided in the data (as we assume) or computed using, e.g., masking or segmentation algorithms. We would like to consider the following indicator function identifying whether a certain pixel is occupied by the rendered object (remember we assume some fixed pixel $p$):

$$S(\theta, \tau) = \begin{cases} 1 & R(\tau) \cap \mathcal{S}_\theta \neq \emptyset \\ 0 & \text{otherwise} \end{cases}$$

Since this function is not differentiable nor continuous in $\theta, \tau$ we use an almost everywhere differentiable approximation:

$$S_\alpha(\theta, \tau) = \text{sigmoid}\left(-\alpha \min_{t \geq 0} f(\boldsymbol{c} + t\boldsymbol{v}; \theta)\right), \tag{7}$$

where $\alpha > 0$ is a parameter. Since, by convention, $f < 0$ inside our geometry and $f > 0$ outside, it can be verified that $S_\alpha(\theta, \tau) \xrightarrow{\alpha \to \infty} S(\theta, \tau)$. Note that differentiating equation 7 w.r.t. $\boldsymbol{c}, \boldsymbol{v}$ can be done using the envelope theorem, namely $\partial_{\boldsymbol{c}} \min_{t \geq 0} f(\boldsymbol{c} + t\boldsymbol{v}; \theta) = \partial_{\boldsymbol{c}} f(\boldsymbol{c} + t_*\boldsymbol{v}; \theta)$, where $t_*$ is an argument achieving the minimum, i.e., $f(\boldsymbol{c}_0 + t_* \boldsymbol{v}_0; \theta) = \min_{t \geq 0} f(\boldsymbol{c}_0 + t\boldsymbol{v}_0; \theta)$, and similarly for $\partial_{\boldsymbol{v}}$. We therefore implement $S_\alpha$ as the neural network $\text{sigmoid}(-\alpha f(\boldsymbol{c} + t_*\boldsymbol{v}; \theta))$. Note that this neural network has exact value and first derivatives at $\boldsymbol{c} = \boldsymbol{c}_0$, and $\boldsymbol{v} = \boldsymbol{v}_0$.

### 3.4 Loss

Let $I_p \in [0, 1]^3$, $O_p \in \{0, 1\}$ be the RGB and mask values (resp.) corresponding to a pixel $p$ in an image taken with camera $\boldsymbol{c}_p(\tau)$ and direction $\boldsymbol{v}_p(\tau)$ where $p \in P$ indexes all pixels in the input collection of images, and $\tau \in \mathbb{R}^k$ represents the parameters of all the cameras in scene. Our loss function has the form:

$$\text{loss}(\theta, \gamma, \tau) = \text{loss}_{\text{RGB}}(\theta, \gamma, \tau) + \rho\,\text{loss}_{\text{MASK}}(\theta, \tau) + \lambda\,\text{loss}_{\text{E}}(\theta) \tag{8}$$

We train this loss on mini-batches of pixels in $P$; for keeping notations simple we denote by $P$ the current mini-batch. For each $p \in P$ we use the sphere-tracing algorithm [12, 17] to compute the first intersection point, $\boldsymbol{c}_p + t_{p,0}\boldsymbol{v}_p$, of the ray $R_p(\tau)$ and $\mathcal{S}_\theta$. Let $P^{\text{in}} \subset P$ be the subset of pixels $p$ where intersection has been found and $O_p = 1$. Let $L_p(\theta, \gamma, \tau) = M(\hat{\boldsymbol{x}}_p, \hat{\boldsymbol{n}}_p, \hat{\boldsymbol{z}}_p, \boldsymbol{v}_p; \gamma)$, where $\hat{\boldsymbol{x}}_p, \hat{\boldsymbol{n}}_p$ is defined as in equations 3 and 4, and $\hat{\boldsymbol{z}}_p = \hat{\boldsymbol{z}}(\hat{\boldsymbol{x}}_p; \theta)$ as in section 3.2 and equation 2. The RGB loss is

$$\text{loss}_{\text{RGB}}(\theta, \gamma, \tau) = \frac{1}{|P|} \sum_{p \in P^{\text{in}}} |I_p - L_p(\theta, \gamma, \tau)|, \tag{9}$$

where $|\cdot|$ represents the $L_1$ norm. Let $P^{\text{out}} = P \setminus P^{\text{in}}$ denote the indices in the mini-batch for which no ray-geometry intersection or $O_p = 0$. The mask loss is

$$\text{loss}_{\text{MASK}}(\theta, \tau) = \frac{1}{\alpha|P|} \sum_{p \in P^{\text{out}}} \text{CE}(O_p, S_{p,\alpha}(\theta, \tau)), \tag{10}$$

Fixed cameras                                   Trained cameras

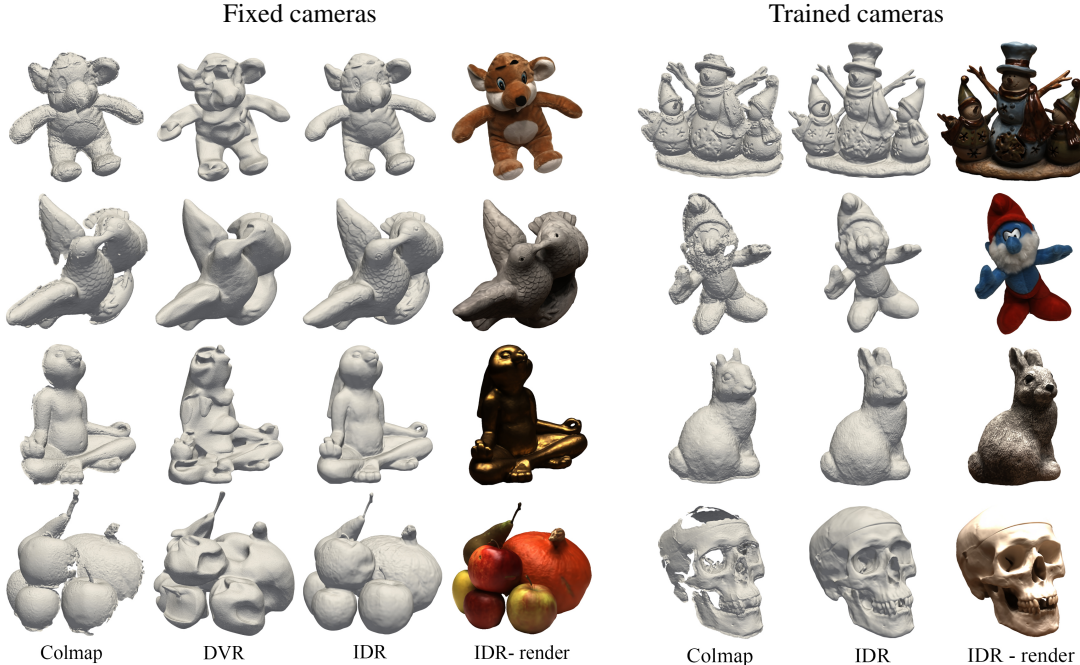

Colmap          DVR          IDR       IDR- render        Colmap          IDR      IDR - render

Figure 4: Qualitative results of multiview 3D surface reconstructions. Note the high fidelity of the IDR reconstructions and its realistic rendering.

where CE is the cross-entropy loss. Lastly, we enforce $f$ to be approximately a signed distance function with Implicit Geometric Regularization (IGR) [11], i.e., incorporating the Eikonal regularization:

$$\text{loss}_\text{E}(\theta) = \mathbb{E}_{\boldsymbol{x}} \big( \|\nabla_{\boldsymbol{x}} f(\boldsymbol{x}; \theta)\| - 1 \big)^2 \tag{11}$$

where $\boldsymbol{x}$ is distributed uniformly in a bounding box of the scene.

**Implementation details.** The MLP $F(\boldsymbol{x}; \theta) = (f(\boldsymbol{x}; \theta), \boldsymbol{z}(\boldsymbol{x}; \theta)) \in \mathbb{R} \times \mathbb{R}^{256}$ consists of 8 layers with hidden layers of width 512, and a single skip connection from the input to the middle layer as in [35]. We initialize the weights $\theta \in \mathbb{R}^m$ as in [2], so that $f(\boldsymbol{x}, \theta)$ produces an approximate SDF of a unit sphere. The renderer MLP, $M(\hat{\boldsymbol{x}}, \hat{\boldsymbol{n}}, \hat{\boldsymbol{z}}, \boldsymbol{v}; \gamma) \in \mathbb{R}^3$, consists of 4 layers, with hidden layers of width 512. We use the non-linear maps of [32] to improve the learning of high-frequencies, which are otherwise difficult to train for due to the inherent low frequency bias of neural networks [37]. Specifically, for a scalar $y \in \mathbb{R}$ we denote by $\boldsymbol{\delta}_k(y) \in \mathbb{R}^{2k}$ the vector of real and imaginary parts of $\exp(i2^\omega \pi y)$ with $\omega \in [k]$, and for a vector $\boldsymbol{y}$ we denote by $\boldsymbol{\delta}_k(\boldsymbol{y})$ the concatenation of $\boldsymbol{\delta}_k(y_i)$ for all the entries of $\boldsymbol{y}$. We redefine $F$ to obtain $\boldsymbol{\delta}_6(\boldsymbol{x})$ as input, i.e., $F(\boldsymbol{\delta}_6(\boldsymbol{x}); \theta)$, and likewise we redefine $M$ to receive $\boldsymbol{\delta}_4(\boldsymbol{v})$, i.e., $M(\hat{\boldsymbol{x}}, \hat{\boldsymbol{n}}, \hat{\boldsymbol{z}}, \boldsymbol{\delta}_4(\boldsymbol{v}); \gamma)$. For the loss, equation 8, we set $\lambda = 0.1$ and $\rho = 100$. To approximate the indicator function with $S_\alpha(\theta, \tau)$, during training, we gradually increase $\alpha$ and by this constrain the shape boundaries in a coarse to fine manner: we start with $\alpha = 50$ and multiply it by a factor of 2 every 250 epochs (up to a total of 5 multiplications). The gradients in equations (11),(4) are implemented using using auto-differentiation. More details are in the supplementary.

## 4 Experiments

### 4.1 Multiview 3D reconstruction

We apply our multiview surface reconstruction model to real 2D images from the DTU MVS repository [15]. Our experiments were run on 15 challenging scans, each includes either 49 or 64 high resolution images of objects with a wide variety of materials and shapes. The dataset also contains ground truth 3D geometries and camera poses. We manually annotated binary masks for all 15 scans except for scans 65, 106 and 118 which are supplied by [33].

We used our method to generate 3D reconstructions in two different setups: (1) fixed ground truth cameras, and (2) trainable cameras with noisy initializations obtained with the linear method of [16]. In both cases we re-normalize the cameras so that their visual hulls are contained in the unit sphere.

| Scan | Trimmed Mesh | | | | Watertight Mesh | | | | | | | |
|---|---|---|---|---|---|---|---|---|---|---|---|---|
| | Colmap$_{trim=7}$ | | Furu$_{trim=7}$ | | Colmap$_{trim=0}$ | | Furu$_{trim=0}$ | | DVR [33] | | IDR | |
| | Chamfer | PSNR | Chamfer | PSNR | Chamfer | PSNR | Chamfer | PSNR | Chamfer | PSNR | Chamfer | PSNR |
| 24 | **0.45** | **19.8** | 0.51 | 19.2 | **0.81** | 20.28 | 0.85 | 20.35 | 4.10(4.24) | 16.23(15.66) | 1.63 | **23.29** |
| 37 | **0.91** | **15.49** | 1.03 | 14.91 | 2.05 | 15.5 | **1.87** | 14.86 | 4.54(4.33) | 13.93(14.47) | 1.87 | **21.36** |
| 40 | **0.37** | **20.48** | 0.44 | 19.18 | 0.73 | 20.71 | 0.96 | 20.46 | 4.24(3.27) | 18.15(19.45) | 0.63 | **24.39** |
| 55 | **0.37** | 20.18 | 0.4 | **20.92** | 1.22 | 20.76 | 1.10 | 21.36 | 2.61(0.88) | 17.14(18.47) | 0.48 | **22.96** |
| 63 | **0.9** | **17.05** | 1.28 | 15.41 | 1.79 | 20.57 | 2.08 | 16.75 | 4.34(3.42) | 17.84(18.42) | 1.04 | **23.22** |
| 65 | **1.0** | **14.98** | 1.22 | 13.09 | 1.58 | 14.54 | 2.06 | 13.53 | 2.81(1.04) | 17.23(20.42) | 0.79 | **23.94** |
| 69 | **0.54** | 18.56 | 0.72 | **18.77** | 1.02 | 21.89 | 1.11 | **21.62** | 2.53(1.37) | 16.33(16.78) | 0.77 | 20.34 |
| 83 | **1.22** | **18.91** | 1.61 | 16.58 | 3.05 | **23.2** | 2.97 | 20.06 | 2.93(2.51) | 18.1(19.01) | 1.33 | 21.87 |
| 97 | **1.08** | 12.18 | 1.37 | **12.36** | 1.4 | 18.48 | 1.63 | 18.32 | 3.03(2.42) | 16.61(16.66) | 1.16 | **22.95** |
| 105 | **0.64** | **20.48** | 0.83 | 19.68 | 2.05 | 21.3 | 1.88 | 20.21 | 3.24(2.42) | 18.39(19.19) | 0.76 | 22.71 |
| 106 | **0.48** | 15.76 | 0.70 | **16.28** | 1.0 | 22.33 | 1.39 | 22.64 | 2.51(1.18) | 17.39(18.1) | 0.67 | **22.81** |
| 110 | **0.59** | **16.71** | 0.87 | 16.53 | 1.32 | 18.25 | 1.45 | 17.88 | 4.80(4.32) | 14.43(15.4) | 0.9 | **21.26** |
| 114 | **0.32** | **19.9** | 0.42 | 19.69 | 0.49 | 20.28 | 0.69 | 20.09 | 4.39(1.04) | 17.08(20.86) | 0.42 | **25.35** |
| 118 | **0.45** | 23.21 | 0.59 | **24.68** | 0.78 | 25.39 | 1.10 | 26.02 | 1.63(0.91) | 19.08(19.5) | 0.51 | 23.54 |
| 122 | **0.43** | 24.48 | 0.53 | **25.64** | 1.17 | 25.29 | 1.16 | 25.95 | 1.58(0.84) | 21.03(22.51) | 0.53 | **27.98** |
| **Mean** | **0.65** | **18.54** | 0.84 | 18.19 | 1.36 | 20.58 | 1.49 | 20.01 | 3.20(2.28) | 17.26(18.33) | **0.9** | **23.20** |

Table 1: Multiview 3D reconstruction with *fixed* cameras, quantitative results for DTU dataset. For DVR we also present (in parentheses) the results with a partial set of images (with reduced reflectance) as suggested in [33].

Training each multi-view image collection proceeded iteratively. Each iteration we randomly sampled 2048 pixel from each image and derived their per-pixel information, including $(I_p, O_p, \boldsymbol{c}_p, \boldsymbol{v}_p), p \in P$. We then optimized the loss in equation 8 to find the geometry $\mathcal{S}_\theta$ and renderer network $M$. After training, we used the Marching Cubes algorithm [29] to retrieve the reconstructed surface from $f$.

**Evaluation.** We evaluated the quality of our 3D surface reconstructions using the formal surface evaluation script of the DTU dataset, which measures the standard Chamfer-$L_1$ distance between the ground truth and the reconstruction. We also report PSNR of train image reconstructions. We note that the ground truth geometry in the dataset has some noise, does not include watertight surfaces, and often suffers from notable missing parts, e.g., Figure 5 and Fig.7c of [33]. We compare to the following baselines: DVR [33] (for fixed cameras), Colmap [40] (for fixed and trained cameras) and Furu [9] (for fixed cameras). Similar to [33], for a fair comparison we cleaned the point clouds of Colmap and Furu using the input masks before running the Screened Poisson Surface Reconstruction (sPSR) [22] to get a watertight surface reconstruction. For completeness we also report their trimmed reconstructions obtained with the trim7 configuration of sPSR that contain large missing parts (see Fig. 5 middle) but performs well in terms of the Chamfer distance.

Quantitative results of the experiment with known fixed cameras are presented in Table 1, and qualitative results are in Figure 4 (left). Our model outperforms the baselines in the PSNR metric, and in the Chamfer metric, for watertight surface reconstructions. In Table 3 we compare the reconstructions obtained with unknown trained camera. Qualitative results for this setup are shown in Figure 4 (right). The relevant baseline here is the Colmap SFM [39]+MVS[40] pipeline.

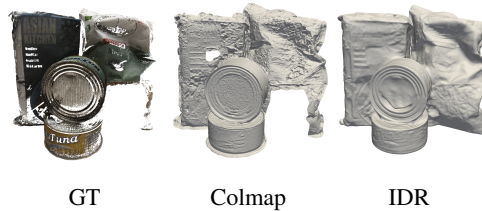

GT　　　　Colmap　　　　IDR

Figure 5: Example of ground truth data (left).

In Figure 7 we further show the convergence of our cameras (rotation and translation errors sorted from small to large) from the initialization of [16] during training epochs along with Colmap's cameras. We note that our method simultaneously improves the cameras parameters while reconstructing accurate 3D surfaces, still outperforming the baselines for watertight reconstruction and PSNR in most cases; scan 97 is a failure case of our method. As can be seen in Figure 4, our 3D surface reconstruction are more complete with better signal to noise ratio than the baselines, while our renderings (right column in each part) are close to realistic.

**Small number of cameras.** We further tested our method on the Fountain-P11 image collections [43] provided with 11 high resolution images with associated GT camera parameters. In Table 2 we show a comparison to Colmap (trim7-sPSR) in a setup of unknown cameras (our method is roughly initialized with [16]). Note the considerable improvement in final camera accuracy over Colmap. Qualitative results are shown in Figure 6.

| | $R_{\text{error}}$(deg) | $t_{\text{error}}$(mm) | PSNR |
|---|---|---|---|
| Colmap | 0.03 | 2.86 | 21.99 |
| IDR | **0.02** | **2.02** | **26.48** |

Table 2: Fountain dataset: cameras accuracy and rendering quality.

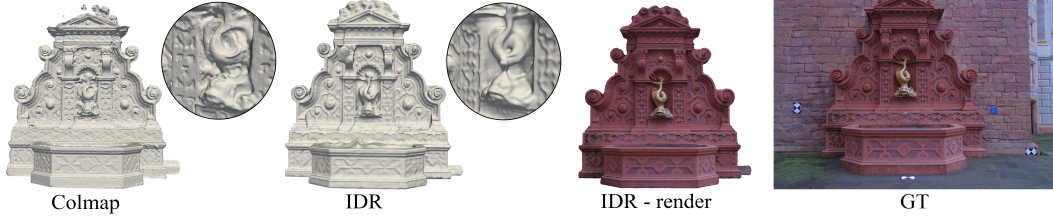

Figure 6: Qualitative results for Fountain data set.

| Scan | Trimmed Mesh | | Watertight Mesh | | | |
|------|--------------|--|-----------------|--|--|--|
| | Colmap$_{trim=7}$ | | Colmap$_{trim=0}$ | | IDR | |
| | Chamfer | PSNR | Chamfer | PSNR | Chamfer | PSNR |
| 24 | 0.38 | 20.0 | **0.73** | 20.46 | 1.96 | **23.16** |
| 37 | 0.83 | 15.5 | **1.96** | 15.51 | 2.92 | **20.39** |
| 40 | 0.3 | 20.67 | **0.67** | 20.86 | 0.7 | **24.45** |
| 55 | 0.39 | 20.71 | 1.17 | 21.22 | **0.4** | **23.57** |
| 63 | 0.99 | 17.37 | 1.8 | 20.67 | **1.19** | **24.97** |
| 65 | 1.45 | 15.2 | 1.61 | 14.59 | **0.77** | **22.6** |
| 69 | 0.55 | 18.5 | 1.03 | 21.93 | **0.75** | **22.91** |
| 83 | 1.21 | 19.08 | 3.07 | **23.43** | **1.42** | 21.97 |
| 97 | 1.03 | 12.25 | 1.37 | **18.67** | - | - |
| 105 | 0.61 | 20.38 | 2.03 | 21.22 | **0.96** | **22.98** |
| 106 | 0.48 | 15.78 | 0.93 | **22.23** | **0.65** | 21.18 |
| 110 | 1.33 | 18.14 | **1.53** | 18.28 | 2.84 | **18.65** |
| 114 | 0.29 | 19.83 | **0.46** | 20.25 | 0.51 | **25.19** |
| 118 | 0.42 | 23.22 | 0.74 | **25.42** | **0.50** | 22.58 |
| 122 | 0.4 | 24.67 | 1.17 | **25.44** | **0.62** | 24.42 |
| **Mean** | 0.71 | 18.75 | 1.35 | 20.68 | **1.16** | **22.79** |

Table 3: Multiview 3D reconstruction with *trained* cameras, quantitative results for DTU dataset.

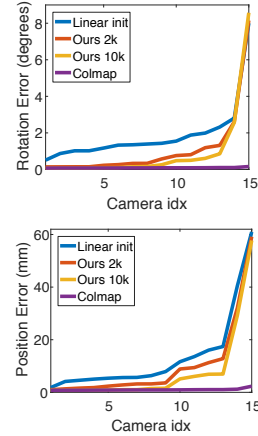

Figure 7: Cameras convergence for the DTU data set.

## 4.2 Disentangling geometry and appearance

To support the claim regarding the necessity of incorporating the surface normal, $n$, in the neural renderer, $M$, for disentangling geometry and appearance (see Section 3.2), we performed the following experiment: We trained two IDR models, each on a different DTU scene. Figure 8 (right) shows (from left to right): the reconstructed geometry; novel views produced with the trained renderer; and novel views produced after switching the renderers of the two IDR models. Figure 8 (left) shows the results of an identical experiment, however this time renderers are not provided with normal input, i.e., $M(\hat{x}, v, \hat{z}; \gamma)$. Note that using normals in the renderer provides a better geometry-appearance separation: an improved surface geometry approximation, as well as correct rendering of different geometries. Figure 9 depicts other combinations of appearance and geometry, where we render novel views of the geometry networks, $f$, using renderers, $M$, trained on different scenes.

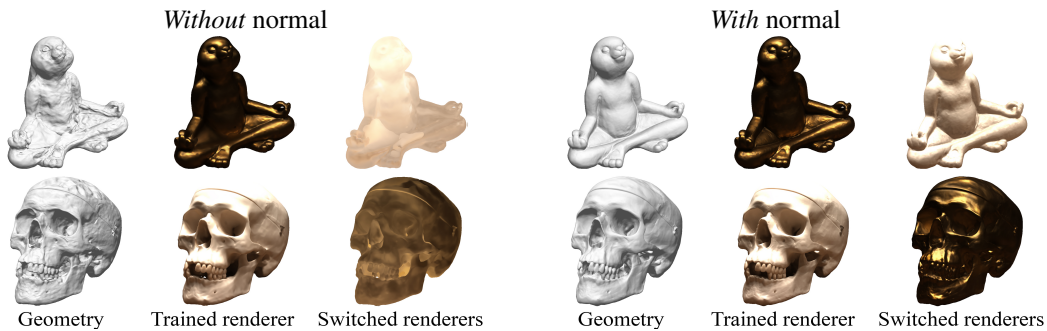

Figure 8: Incorporating normal in the renderer allows accurate geometry and appearance disentanglement: comparison of reconstructed geometry, novel view with trained renderer, and novel view with a different scene's renderer shown with and without normal incorporated in the renderer.

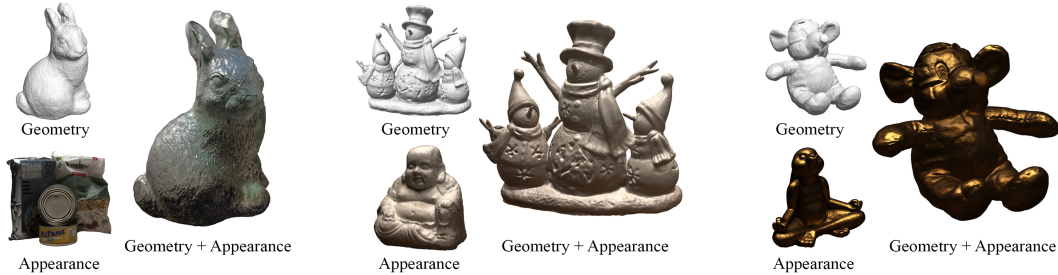

Figure 9: Transferring appearance to unseen geometry.

## 4.3 Ablation study

We used scan 114 of the DTU dataset to conduct an ablation study, where we removed various components of our renderer $M(\hat{\boldsymbol{x}}, \hat{\boldsymbol{n}}, \hat{\boldsymbol{z}}, \boldsymbol{v}; \gamma)$, including (see Figure 10): (a) the viewing direction $\boldsymbol{v}$; (b) the normal $\hat{\boldsymbol{n}}$; and (c) the feature vector, $\hat{\boldsymbol{z}}$. (d) shows the result with the full blown renderer $M$, achieving high detailed reconstruction of this marble stone figure (notice the cracks and fine details). In contrast, when the viewing direction, normal, or feature vector are removed the model tends to confuse lighting and geometry, which often leads to a deteriorated reconstruction quality.

In Figure 10 (e) we show the result of IDR training with fixed cameras set to the inaccurate camera initializations obtained with [16]; (f) shows IDR results when camera optimization is turned on. This indicates that the optimization of camera parameters together with the 3D geometry reconstruction is indeed significant.

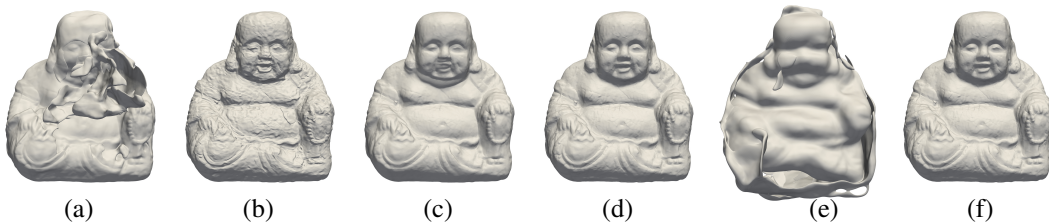

| (a) | (b) | (c) | (d) | (e) | (f) |

Figure 10: Ablation study, see text for details.

## 5 Conclusions

We have introduced the Implicit Differentiable Renderer (IDR), an end-to-end neural system that can learn 3D geometry, appearance, and cameras from masked 2D images and noisy camera intializations. Considering only rough camera estimates allows for robust 3D reconstruction in realistic scenarios in which exact camera information is not available. One limitation of our method is that it requires a reasonable camera initialization and cannot work with, say random camera initialization. Interesting future work is to combine IDR with a neural network that predicts camera information directly from the images. Another interesting future work is to further factor the surface light field ($M_0$ in equation 5) into material (BRDF, $B$) and light in the scene ($L^i$). Lastly, we would like to incorporate IDR in other computer vision and learning applications such as 3D model generation, and learning 3D models from images in the wild.

## Acknowledgments

LY, MA and YL were supported by the European Research Council (ERC Consolidator Grant, "LiftMatch" 771136) and the Israel Science Foundation (Grant No. 1830/17). YK, DM, MG and RB were supported by the U.S.- Israel Binational Science Foundation, grant number 2018680 and by the Kahn foundation. The research was supported also in part by a research grant from the Carolito Stiftung (WAIC).

## Broader Impact

In our work we want to learn 3D geometry of the world from the abundant data of 2D images. In particular we allow high quality of 3D reconstruction of objects and scenes using only standard images. Applications of our algorithm could be anywhere 3D information is required, but only 2D images are available. This could be the case in: product design, entertainment, security, medical imaging, and more.

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
