[Supplementary Material · 2020_IDR (5).pdf]

# Multiview Neural Surface Reconstruction by Disentangling Geometry and Appearance
## Supplementary material

**Lior Yariv**    **Yoni Kasten**    **Dror Moran**

**Meirav Galun**    **Matan Atzmon**    **Ronen Basri**    **Yaron Lipman**

Weizmann Institute of Science

{lior.yariv, yoni.kasten, dror.moran, meirav.galun, matan.atzmon, ronen.basri, yaron.lipman}@weizmann.ac.il

## 1    Additional implementation details

### 1.1    Training details

We trained our networks using the ADAM optimizer [10] with a learning rate of $1e-4$ that we decrease by a factor of 2 at epochs 1000 and 1500. Each model was trained for $2K$ epochs. Training was done on a single Nvidia V-100 GPU, using PYTORCH deep learning framework [17]. Training time are 6.5 or 8 hours for 49 or 64 images, respectively. Rendering (inference) time: 30 seconds for $1200 \times 1600$ image with 100K pixel batches.

**Images preprocessing.** As in [21] we transform the RGB images such that each pixel is in the range $[-1, 1]^3$ by subtracting 0.5 and multiplying by 2.

### 1.2    Camera representation and initialization

We represent a camera by $\mathcal{C} = (\boldsymbol{q}, \boldsymbol{c}, \boldsymbol{K})$, where $\boldsymbol{q} \in \mathbb{R}^4$ is a quaternion vector representing the camera rotation, $\boldsymbol{c} \in \mathbb{R}^3$ represents the camera position, and $\boldsymbol{K} \in \mathbb{R}^{3\times3}$ is the camera's intrinsic parameters. Our cameras' parameters are $\tau = (\mathcal{C}_1, \dots, \mathcal{C}_N)$, where $N$ is the numbers of cameras (also images) in the scene. Let $\boldsymbol{Q}(\boldsymbol{q}) \in \mathbb{R}^{3\times3}$ denote the rotation matrix corresponding to the quaternion $\boldsymbol{q}$. Then, for camera $\mathcal{C}_i$, $i \in [N]$, and a pixel $p$ we have:

$$\boldsymbol{c}_p(\tau) = \boldsymbol{c}_i \tag{A1}$$

$$\boldsymbol{v}_p(\tau) = \frac{1}{\left\|\boldsymbol{K}_i^{-1}\boldsymbol{p}\right\|_2}\boldsymbol{Q}(\boldsymbol{q}_i)\boldsymbol{K}_i^{-1}\boldsymbol{p}, \tag{A2}$$

where $\boldsymbol{p} = (p_x, p_y, 1)^T$ is the pixel $p$ in homogeneous coordinates.

In our experiments with unknown cameras, for each scene (corresponding to one 3D object) we generated relative motions between pairs of cameras using SIFT features matching [12] and robust essential matrix estimation (RANSAC), followed by a decomposition to relative rotation and relative translation [6]. We used these relative motions as inputs to the linear method of [9] to produce noisy cameras initialization for our method. We assume known intrinsics, as commonly assumed in calibrated SFM.

### 1.3    Architecture:

Figure A1 visualize the IDR architecture. The gray blocks describe the input vectors, orange blocks for the output vectors, and each blue block describe a fully connected hidden layer with **softplus** activation ; a smooth approximation of **ReLU**: $x \mapsto \frac{1}{\beta}\ln(1 + e^{\beta x})$. We used $\beta = 100$. In the

Figure A1: IDR architecture.

renderer network we used the **ReLU** activation between hidden layers and **tanh** for the output, to get valid color values. The grey block illustrate the use of equation 3.

As mentioned in the main paper, we initialize the weights of the implicit function $\theta \in \mathbb{R}^m$ as in [2], so that $f(\boldsymbol{x}, \theta)$ produces an approximate SDF of a unit sphere. To use the non-linear maps of [14], and still maintain the geometric initialization we implement the positional encoding embedding with the addition of concatenating identity embedding, that is: $\boldsymbol{\delta}'_k(\boldsymbol{y}) = [\boldsymbol{y}, \boldsymbol{\delta}_k(\boldsymbol{y})]$, where $\boldsymbol{\delta}_k(\boldsymbol{y})$ as defined in the section 3.4. Then, we zero out the weights coming out of everything except the linear basis function, to get the proper geometric initialization as in [2].

## 1.4 Ray marching algorithm

We denote by $\boldsymbol{c}$ the camera center and by $\boldsymbol{v} \in \mathcal{S}$ the direction from $\boldsymbol{c}$ associated with a pixel on the image plane. During our training for each selected pixel, we perform ray tracing along the ray $\{\boldsymbol{c} + t\boldsymbol{v} \mid t \geq 0\}$ to find the corresponding first intersection point with $\mathcal{S}_\theta$ denoted by $\boldsymbol{x}_0$. As $\mathcal{S}_\theta$ is defined by the zero level set of an approximated sign distance field $f$, we base our tracing method on the sphere tracing algorithm [5]: in each iteration, since the distance along the ray to $\boldsymbol{x}_0$ is bounded by the current SDF value $f(\boldsymbol{c} + t\boldsymbol{v})$, we march forward by enlarging the current $t$ by $f(\boldsymbol{c} + t\boldsymbol{v})$. We continue iterating until convergence, indicated by an SDF value below a threshold of 5e-5, or divergence, indicated by a step that reaches out of the unit sphere. As we assume the 3D object to be located inside the unit sphere, we start the sphere tracing algorithm at the first intersection point of the ray with the unit sphere as in [11].

We limit the number of sphere tracing steps to 10 in order to prevent long convergence cases e.g., when the ray passes very close to the surface without crossing it. Then, for non convergent rays, we run another 10 sphere tracing iterations starting from the second (i.e., farthest) intersection of the ray with the unit sphere and get $\bar{t}$. Similar to [16] we sample 100 equal sized steps between $t$ to $\bar{t}$: $t < t_1 < \cdots < t_{100} < \bar{t}$ and find the first pair of consecutive steps $t_i, t_{i+1}$ such that $\text{sign}(f(\boldsymbol{c} + t_i\boldsymbol{v})) \neq \text{sign}(f(\boldsymbol{c} + t_{i+1}\boldsymbol{v}))$, i.e., a sign transition that describes the first ray crossing of the surface. Then, using $t_i$ and $t_j$ as initial values, we run 8 iterations of the secant algorithm to find an approximation for the intersection point $\boldsymbol{x}_0$. The algorithm is performed in parallel on the GPU for multiple rays associated with multiple pixels.

For all pixels in $P^{\text{out}}$ we select a point on the corresponding ray $\boldsymbol{c} + t_*\boldsymbol{v}$ for $\text{loss}_{\text{MASK}}$, where $t_*$ is approximated by sampling uniformly 100 possible steps $\mathcal{T} = \{t_1, ..., t_{100}\}$ between the two intersection points of the ray with the unit sphere and taking $t_* = \arg\min_{t_i \in \mathcal{T}} f(\boldsymbol{c} + t_i\boldsymbol{v})$.

All the 3D points resulted from the described process are used for equation 11 together with additionally 1024 points distributed uniformly in the bounding box of the scene.

## 1.5 Baselines methods running details

In our experiments we compared our method to Colmap [18],[19], DVR [16] and Furu [4]. We next describe implementation details for these baselines.

**Colmap.** We used the official Colmap implementation [20]. For unknown cameras, only the intrinsic camera parameters are given, and we used the "mapper" module to extract camera poses. For fixed known cameras the GT poses are given as inputs. For both setups we run their "feature_extractor", "exhaustive_matcher" , "point_triangulator", "patch_match_stereo" and "stereo_fusion" modules to generate point clouds. We also used their screened Poisson Surface Reconstruction (sPSR) for 3D mesh generation after cleaning the point clouds as described in Section 4.1. For rendering, we used their generated 3D meshes and cameras, and rendered images for each view using the "Pyrender" package [13].

**DVR.** We run DVR using the official Github code release [15]. In order to be compatible with this implementation, for each scene we used the cameras and the masks to normalize all the cameras such that the object is inside the unit cube. We applied DVR on all DTU scenes using the "ours_rgb" configuration. As mentioned in Table 1 (in the main paper), we reconstructed each model twice: with all the images in the dataset, and with all the images in the dataset that are not in the DVR "ignore list". We run their method for 5000 epochs and took the best model as they suggest. We further used their "generate" and "render" scripts for generating 3D meshes and rendering images respectively.

**Furu.** The point clouds generated by Furu are supplied by the DTU data set. We used Colmap's sPSR to generate meshes from the cleaned point clouds. As for Colmap we used "Pyrender" [13] to render their images.

## 1.6 Evaluation details

All the reported Chamfer distances are in millimeters and computed by averaging the accuracy and completeness measures, returned from DTU evaluation script, each represents one sided Chamfer-$L_1$ distance. The reported PSNR values (in dB) for each scan are averaged over the masked pixels of all the images. Camera accuracy represents the mean accuracy of camera positions (in mm) and camera rotations (in degrees) after L1 alignment with the GT.

## 2 Additional results

### 2.1 Multiview 3D reconstruction

In Figure A2 we show a failure case (scan 37 from the DTU dataset). This case presents challenging thin metals parts which our method fails to fully reproduce especially with noisy cameras.

In Figure A3 we present qualitative results for 3D surface reconstruction with fixed GT cameras, whereas Figure A4 further presents qualitative results for the unknown cameras setup. An ablation study for training IDR with fixed noisy cameras compared to IDR's camera training mode initialized with the same noisy cameras is also presented in Figure A4. We refer the reader to a supplementary video named "IDR.mp4" that shows side by side our generated surfaces and rendered images from multiple views in the settings of unknown cameras.

Fixed cameras

Trained cameras

Figure A2: Failure cases.

**Unexpected lighting effects in the supplied video.** In the original views (training images), the camera body occasionally occludes some light sources, casting temporary shadows on the object; see inset. Notice that in the video we move the camera (i.e., viewing direction) and not the object, therefore when the camera moves near such a view we see the projected shadow in the generated rendering.

### 2.2 Disentangling geometry and appearance

Figure A5 depicts other transformation of appearance to unseen geometry, where we render novel views of different trained geometries, using the renderer from the model trained on the golden bunny scene.

Figure A3: Qualitative results of multiview 3D surface reconstructions with fixed GT cameras for the DTU dataset. We present surface reconstructions generated by our method compared to the baseline methods. Our renderings from two novel views are presented as well.

## 2.3 PSNR on a held-out set of images

In our experiments the PSNR is computed over training images. We further performed an experiment where we held-out $10\%$ of the images as test views, when using fixed GT cameras. Overall, we got $23.34$ / $22.55$ mean PSNR accuracy on the train / test images, respectively. Table 1 reports the per-scene accuracies.

| Scan | 24 | 37 | 40 | 55 | 63 | 65 | 69 | 83 | 97 | 105 | 106 | 110 | 114 | 118 | 122 | Mean |
|---|---|---|---|---|---|---|---|---|---|---|---|---|---|---|---|---|
| **PSNR**(Test) | 23.02 | 19.61 | 24.07 | 22.30 | 23.43 | 22.30 | 22.19 | 21.70 | 22.05 | 23.82 | 21.75 | 20.31 | 23.03 | 23.42 | 25.31 | 22.55 |
| **PSNR**(Train) | 24.13 | 21.02 | 24.65 | 23.51 | 24.81 | 23.33 | 22.74 | 21.80 | 23.29 | 24.36 | 22.04 | 20.32 | 23.89 | 23.72 | 26.45 | 23.34 |

Table 1: PSNR on a held-out set of images with fixed GT cameras.

Figure A4: Qualitative results for trained cameras setup. We also compare to IDR with fixed noisy cameras.

## 3 Differentiable intersection of viewing direction and geometry (proof of Lemma 1)

We will use the notation of the main paper, repeated here for convenience: $\hat{\boldsymbol{x}}(\theta, \tau) = \boldsymbol{c}(\tau) + t(\theta, \boldsymbol{c}(\tau), \boldsymbol{v}(\tau))\boldsymbol{v}(\tau)$ denotes the first intersection point of the viewing ray $R_p(\tau)$ for some fixed pixel $p$ and the geometry $\mathcal{S}_\theta$. We denote the current parameters by $\theta_0, \tau_0$; accordingly we denote $\boldsymbol{c}_0 = \boldsymbol{c}(\tau_0)$, $\boldsymbol{v}_0 = \boldsymbol{v}(\tau_0)$, and $t_0 = t(\theta_0, \boldsymbol{c}_0, \boldsymbol{v}_0)$. We also denote $\boldsymbol{x}_0 = \hat{\boldsymbol{x}}(\theta_0, \tau_0)$. Note that $\boldsymbol{x}_0 = \boldsymbol{c}_0 + t_0\boldsymbol{v}_0$ is the intersection point at the current parameters that is $R_p(\tau_0) \cap \mathcal{S}_{\theta_0}$.

To find the function dependence of $\hat{\boldsymbol{x}}$ on $\theta, \tau$ we use implicit differentiation [1, 16]. That is we differentiate

$$f(\hat{\boldsymbol{x}}(\theta, \tau); \theta) \equiv 0 \qquad (A3)$$

w.r.t. its parameters. Since the functional dependence of $\boldsymbol{c}$ and $\boldsymbol{v}$ on $\tau$ is known (given in equations A1-A2) it is sufficient to consider the derivatives of equation A3 w.r.t. $\theta, \boldsymbol{c}, \boldsymbol{v}$. Therefore we consider

$$f(\boldsymbol{c} + t(\theta, \boldsymbol{c}, \boldsymbol{v})\boldsymbol{v}; \theta) \equiv 0 \qquad (A4)$$

and differentiate w.r.t. $\theta, c, v$. We differentiate w.r.t. $c$: (note that we use $\partial f/\partial x$ equivalently to $\nabla_x f$, which is used in the main paper)

$$\left(\frac{\partial f}{\partial x}\right)^T \left(I + v \left(\frac{\partial t}{\partial c}\right)^T\right) = 0,$$

where $I \in \mathbb{R}^{3\times 3}$ is the identity matrix, and $v, \frac{\partial f}{\partial x}, \frac{\partial t}{\partial c} \in \mathbb{R}^3$ are column vectors. Rearranging and evaluating at $\theta_0, c_0, v_0$ we get

$$\frac{\partial t}{\partial c}(\theta_0, c_0, v_0) = -\frac{1}{\left\langle \frac{\partial f}{\partial x}(x_0; \theta_0), v_0 \right\rangle} \frac{\partial f}{\partial x}(x_0; \theta_0). \tag{A5}$$

Differentiating w.r.t. $v$:

$$\left(\frac{\partial f}{\partial x}\right)^T \left(tI + v \left(\frac{\partial t}{\partial v}\right)^T\right) = 0,$$

Rearranging and evaluating at $\theta_0, c_0, v_0$ we get

$$\frac{\partial t}{\partial v}(\theta_0, c_0, v_0) = -\frac{t_0}{\left\langle \frac{\partial f}{\partial x}(x_0; \theta_0), v_0 \right\rangle} \frac{\partial f}{\partial x}(x_0; \theta_0). \tag{A6}$$

Lastly, the derivatives w.r.t. $\theta$ are as in [16] and we give it here for completeness:

$$\left(\frac{\partial f}{\partial x}\right)^T v \frac{\partial t}{\partial \theta} + \frac{\partial f}{\partial \theta} = 0.$$

Reordering and evaluating at $\theta_0, v_0, c_0$ we get

$$\frac{\partial t}{\partial \theta}(\theta_0, c_0, v_0) = -\frac{1}{\left\langle \frac{\partial f}{\partial x}(x_0; \theta_0), v_0 \right\rangle} \frac{\partial f}{\partial \theta}(x_0; \theta_0) \tag{A7}$$

Now consider the formula

$$t(\theta, c, v) = t_0 - \frac{1}{\left\langle \frac{\partial f}{\partial x}(x_0; \theta_0), v_0 \right\rangle} f(c + t_0 v; \theta). \tag{A8}$$

Evaluating at $\theta_0, c_0, v_0$ we get $t(\theta_0, c_0, v_0) = t_0$ since $f(x_0; \theta_0) = 0$. Furthermore, differentiating $t$ w.r.t. $c, v, \theta$ and evaluating at $c_0, v_0, \theta_0$ we get the same derivatives as in equations A5, A6, and A7. Plugging equation A8 into $\hat{x}(\theta, c, v) = c + t(\theta, c, v)v$ we get the formula,

$$\hat{x}(\theta, \tau) = c + t_0 v - \frac{v}{\left\langle \frac{\partial f}{\partial x}(x_0; \theta_0), v_0 \right\rangle} f(c + t_0 v; \theta), \tag{A9}$$

that coincide in value and first derivatives with the first intersection points at the current parameters $\theta_0, c_0, v_0$; equation A9 can be implemented directly by adding a linear layer in the entrance to the network $f$ and a linear layer at its output.

Figure A5: Transferring appearance from scan 110 (golden bunny) to unseen geometry.

# 4 Restricted surface light field model $\mathcal{P}$

We restrict our attention to light fields that can be represented by a *continuous* function $M_0$. We denote the collection of such continuous functions by $\mathcal{P} = \{M_0\}$.

This model includes many common materials and lighting conditions such as the popular Phong model [3]:

$$L(\hat{\boldsymbol{x}}, \boldsymbol{w}^o) = k_d O_d I_a + k_d O_d I_d \left( \hat{\boldsymbol{n}} \cdot \frac{\boldsymbol{\ell} - \hat{\boldsymbol{x}}}{\|\boldsymbol{\ell} - \hat{\boldsymbol{x}}\|} \right)_+ + k_s O_s I_d \left( \hat{\boldsymbol{r}} \cdot \frac{\boldsymbol{\ell} - \hat{\boldsymbol{x}}}{\|\boldsymbol{\ell} - \hat{\boldsymbol{x}}\|} \right)_+^{n_s}, \qquad (A10)$$

where $(a)_+ = \max\{a, 0\}$; $k_d, k_s$ are the diffuse and specular coefficients, $I_a, I_d$ are the ambient and point light source colors, $O_d, O_s$ are the diffuse and specular colors of the surface, $\boldsymbol{\ell} \in \mathbb{R}^3$ is the location of a point light source, $\hat{\boldsymbol{r}} = -(\boldsymbol{I} - 2\hat{\boldsymbol{n}}\hat{\boldsymbol{n}}^T)\boldsymbol{w}^o$ the reflection of the viewing direction $\boldsymbol{w}^o = -\boldsymbol{v}$ with respect to the normal $\hat{\boldsymbol{n}}$, and $n_s$ is the specular exponent.

Note, that the family $\mathcal{P}$, however, does not include all possible lighting conditions. It excludes, for example, self-shadows and second order (or higher) light reflections as $L^i$ is independent of the geometry $\mathcal{S}_\theta$.

## 4.1 $\mathcal{P}$-Universality of renderer

We consider renderer of the form $M(\boldsymbol{x}, \boldsymbol{n}, \boldsymbol{v}; \gamma)$, where $M$ is an MLP, and show it can approximate the correct light field function $M_0(\boldsymbol{x}, \boldsymbol{n}, \boldsymbol{v})$ for all $(\boldsymbol{x}, \boldsymbol{n}, \boldsymbol{v})$ contained in some compact set $\mathcal{K} \subset \mathbb{R}^9$. Indeed, for some arbitrary and fixed $\epsilon > 0$ the universal approximation theorem for MLPs (see [8] and [7] for approximation including derivatives) guarantees the existence of parameters $\gamma \in \mathbb{R}^n$ so that $\max_{(\boldsymbol{x}, \boldsymbol{n}, \boldsymbol{v}) \in \mathcal{K}} |M(\boldsymbol{x}, \boldsymbol{n}, \boldsymbol{v}; \gamma) - M_0(\boldsymbol{x}, \boldsymbol{n}, \boldsymbol{v})| < \epsilon$.

## 4.2 View direction and normal are necessary for $\mathcal{P}$-universality.

We will next prove that taking out $\boldsymbol{v}$ or $\boldsymbol{n}$ from the renderer $M$ will make this model not $\mathcal{P}$-universal. Assume $\mathcal{S}_\theta$ is expressive enough to change the normal direction $\hat{\boldsymbol{n}}$ arbitrarily at a point $\hat{\boldsymbol{x}}$; for example, even a linear classifier $f(\boldsymbol{x}; \boldsymbol{a}, b) = \boldsymbol{a}^T \boldsymbol{x} + b$ would do. We will use the Phong model (equation A10) to prove the claims; the Phong model is in $\mathcal{P}$ as discussed above. We first consider the removal of the normal component $\hat{\boldsymbol{n}}$ from the renderer, i.e., $M(\hat{\boldsymbol{x}}, \boldsymbol{v}; \gamma)$ is the approximated light field radiance arriving at $\boldsymbol{c}$ in direction $\boldsymbol{v}$ as predicted by the renderer. Consider the setting shown in Figure 3 (a) and (b) (in the main paper): In both cases $M(\hat{\boldsymbol{x}}, \boldsymbol{v}; \gamma)$ yields the same value although changing the normal direction at $\hat{\boldsymbol{x}}$ will produce, under the Phong model, a different light amount at $\boldsymbol{c}$.

Similarly, consider a renderer with the viewing direction, $\boldsymbol{v}$, removed, i.e., $M(\hat{\boldsymbol{x}}, \hat{\boldsymbol{n}}; \gamma)$, and Figure 3 (a) and (c): In both cases $M(\hat{\boldsymbol{x}}, \hat{\boldsymbol{n}}; \gamma)$ produces the same value although, under the Phong model, the reflected light can change when the point of view changes. That is, we can choose light position $\boldsymbol{\ell}$ in equation A10 so that different amount of light reaches $\boldsymbol{c}$ at direction $\boldsymbol{v}$.

We have shown that renderers with no access to $\hat{\boldsymbol{n}}$ and/or $\boldsymbol{v}$ would necessarily fail (at-least in some cases) predicting the correct light amount reaching $\boldsymbol{c}$ in direction $\boldsymbol{v}$ when these and the geometry is able to change, hence are not universal.