[Reviews · NeurIPS 2020]

Review 1

Summary and Contributions: This paper presents a method for recovering both the geometry and appearance of an object from many input images. For each object, a network is trained that represents geometry as the zero level set of a signed distance function and outgoing color values as a function of position and direction. Unlike prior work, this method is able to correct for noisy input camera pose estimates as part of per-scene gradient-based optimization procedure. Quantitative and qualitative results on the DTU benchmark demonstrate the high quality of both the geometry and novel rendered views.

Strengths: This is excellent work that combines recent advances in neural scene representations to achieve very impressive 3D reconstruction results. My interpretation is that the success builds on combining: - differentiable implicit surface intersection from DVR [33], - positional encoding and view direction as input from NeRF [32], - principled SDF network training from IGR [11] and SAL [2], along with the unique addition of backpropagating all the way through to the camera pose parameters, which I have not seen before. This is also the first neural scene representation work I know of that reports both geometric reconstruction accuracy (Chamfer) and rerendering accuracy (PSNR) -- it is great to highlight both of these, rather than categorizing the work as purely shape reconstruction or purely view synthesis.

Weaknesses: I recognize the clear contribution of the paper and high quality of the results. However, I find the claims about the IDR module very misleading and cannot recommend acceptance without this being addressed. See “Clarity” for more. More minor criticisms: I thank the authors for being very clear that their work requires manually annotated object masks. However, this does provide an extremely helpful “space carving” type cue for reconstruction, and I wonder how much this boosts the quality (clearly it does not provide all the fine geometry detail, but I’m sure it is helping yield cleaner silhouettes and eliminating some potential floater artifacts). This is one detail that could prevent this method from becoming a new go-to baseline for 3D reconstruction of objects in the same way COLMAP is today since it makes it less plug-and-play. I couldn’t find any mention of speed for either training or inference. How long does it take to train the networks for one scene? How long does it take to render a new view of a scene from a trained network? Things I do not consider negatives but that could potentially add to the paper: Results are only shown on the DTU dataset (plus the Fountain scene). They are certainly convincing, but it would be more thorough to report on other datasets such as Tanks and Temples, the recent BlendedMVS dataset, or particularly on handheld captures of objects taken by the authors themselves. Comparisons are provided against “classic” dense multiview mesh reconstruction methods COLMAP and Furu and only against one more modern method, DVR. I could not find an obvious missing deep learning method that would have allowed for both Chamfer and PSNR comparisons, so I think this is sufficient. It would be interesting to also report PSNR-only comparisons against a state of the art method for view synthesis, to see how they compare.

Correctness: The method and methodology seem correct. However, see "Clarity" for complaints about details that could be considered misleading to the point of being incorrect.

Clarity: As written, I find the explanation of the “implicit differentiable renderer” to be highly misleading. Throughout the paper, the network M is described as a “differentiable renderer” that accounts for both BRDF and lighting conditions. Indeed, the TITLE of the paper implies that lighting and materials are being recovered in the style of full inverse rendering. Section 3.2 introduces the rendering equation and makes a big deal about specifying the BRDF and light sources. However, this is all rendered pointless by lines 155-156: “Replacing M0 with a (sufficiently large) MLP approximation M provides the radiance approximation…” Rolling up all these factors into one big function means you are learning a surface light field, nothing more. There is no ability to disentangle lighting and materials here. No losses are applied during training to encourage this to happen. The authors then proceed to make strange claims about the necessity of viewing direction and normal as inputs, and the need to pass a global feature vector z to account for global lighting effects. This greatly confused me, since the network M is being trained end-to-end along with the surface network. Once M is given x and v (location and view direction) as inputs, it is already arbitrarily powerful and can represent any plenoptic function it wants, which becomes a surface light field when restricted to the level set of F. Passing the normal and the z vector clearly helps based on the ablations, but this is likely because 1) surface color is closely tied to normal vector due to Lambert’s cosine law and the way BRDFs work and 2) z comes out of a network that has a positional encoding on x, whereas positional encoding is not used when passing x to M. I believe M is probably acting more similarly to a renderer due to the fact that the normal is passed in, but you cannot make these claims with no evidence to back it up. I would believe this more if the authors had shown that you could modify the surface after training and still use the same M network to shade it correctly. The logic in supplement section 4.2 is suspect for the same reason: on lines 121-128, the authors write that without the normal as input, M would yield an incorrect value if the surface normal changed (with constant position, lighting, and viewing direction). This doesn’t make sense since F (implying surface S) and M are always trained jointly and never modified separately in the results. Yes, if the normal changed then M wouldn’t know if it wasn’t an input, however this never happens since F and M are jointly modified at every step of the optimization. If the surface normal changes through gradient descent, so will the outgoing radiance returned by M to match. At a higher level than these details, the overall language and title convey that the paper is doing something it is not (recovering separable materials/lighting rather than just a surface light field). Section 3.2 in the main text and Section 4 in the supplement obfuscate this with an unjustified formalism, introducing many terms and functions that in the end have no relation to what is actually being learned. I am reacting particularly strongly to this since NeurIPS readers are less likely to be familiar with these graphics principles and thus more likely to be mislead. I request that the authors propose an alternate title and sketch out a restructuring of section 3.2 in the rebuttal.

Relation to Prior Work: The related work section is thorough and focuses on the large number of recent differentiable ray-casting and neural representation papers. As enumerated in “Strengths,” I think it is clear how this work builds on and adds to these earlier methods.

Reproducibility: No

Additional Feedback: Just by virtue of how many implementation details there are, it would be hard to reimplement from scratch, so I encourage the authors to release code (as well as the DTU image masks) for replicability. ==================== Post-rebuttal feedback: - Thank you for addressing the concerns from various reviewers about misleading claims. The new title would be much clearer for future readers, and I strongly suggest following through on the proposed text changes in section 3.2 for the revised version. - The included result showing the transferability of the trained M renderer network is extremely interesting! I would suggest prominently featuring this in the revised main text. I think this simple visual example will capture peoples' attention and spur further investigation of how to exploit this disentanglement. - Certainly do also update the PSNR to be on test set, not training set, else this might be perceived as a flaw in methodology since reporting rerendering error on held-out images is more standard.


Review 2

Summary and Contributions: The paper presents a novel method for reconstruction of a 3D surface and appearance from multiple views. The surface is represented by a neural network that approximates the signed-distance field of the shape, and the appearance is captured by another neural network that learns a mapping from the position, normal, viewing direction, and a global latent code to a RGB color. The method is evaluated on a standard MVS dataset containing photographs of various small objects such as sculptures and plush toys that have been segmented from their backgrounds, and performs comparably or better than state-of-the-art non-learned MVS in shape reconstruction, while also learning the view-dependent appearance of each shape. The method appears to be focused entirely on surface reconstruction and does not handle volumetric effects.

Strengths: The quality of the results is very impressive. The system manages to complete parts of the shapes that COLMAP cannot reconstruct, such as the eye sockets of the skull shape. The rerendered results look essentially like photographs of the original shape. The results look so good in some cases that I'm not sure you could do better even by spray painting the shapes and using an active-light scanner to capture them. The ability to optimize the camera positions is a nice addition, and the experiments demonstrate that it makes a big difference versus roughly oriented cameras. The method itself is pretty simple, and seems like it should be practical to implement. It is closely related to NeRF and shares the good performance, but NeRF (I believe) counts as concurrent work so that is not a problem for this paper. The method is evaluated on a challenging dataset that includes a variety of shapes and appearances, and overall the protocol is pretty good (but see below). Ablations are provided for the various inputs to the rendering network. [post rebuttal] The material swap results look really good and should definitely be included in the paper.

Weaknesses: While the still images look amazing, watching the video I felt the lighting on the shapes moved in unexpected ways that made me question the accuracy of the lighting reconstruction. From what I can tell about the evaluation protocol, the PSNR was computed over the same images used in training the model. It would be stronger result to show PSNR on a held-out set of images. As it is, it's difficult to evaluate the generalization performance of the model. The term and discussion of "P-universality" seems somewhat misleading. While position, normal, and viewing direction are a natural parameterization for surface rendering, they are not a necessary parameterization for rendering in general; light field rendering just deals with radiance values disconnected to any surface normal (the L^e and L^i at l. 152), but is also general (or "universal"). Hence it doesn't really seem accurate to say that renderers without n^hat are not "universal." There is no discussion in the paper or the supplemental material about running time, either for training or for inference. Slow inference is a weakness for these types of models, so it would be nice to have some sense of how slow we are talking about.

Correctness: Seems correct. Proofs are given for several derivations.

Clarity: The paper is overall well written, though it is very dense (especially the evaluation section). I'm not entirely clear what it means to "add two linear layers with parameters c, v" around the MLP (l. 131) though that detail seems important. It would be helpful to supply code for the model to accompany the paper.

Relation to Prior Work: Seems ok. This is such a hot area that a lot of the references are to arxiv submissions which is messy, but unavoidable I guess.

Reproducibility: Yes

Additional Feedback: While I understand that the protocol for the DTU set is well-defined, F-score might be a more intuitive measurement than Chamfer distance. How were the 15 scans used in the results chosen? A figure or measurement of how misaligned the cameras are in the trainable setting would be nice. Does the camera optimization still matter if bundle-adjustment has already been run on the cameras? The experiments seem to be run with either roughly aligned cameras or GT cameras. It would be interesting to see results with camera poses produced by COLMAP, as that is a more realistic setting (for e.g., casual object scanning with a phone). How often does the sphere tracing fail and require fallback to the brute-force solution? Early in training it seems the SDF criterion cannot be met everywhere. Broader impact does not consider negative consequences of the work, though they seem limited.


Review 3

Summary and Contributions: This paper proposes to estimate object geometry using an implicit representation (regularized by the Eikonal criterion) from masked 2D images. The camera poses are required but can be coarse and refined during the optimization. Another neural network is used to represent the radiance field modeling the object's appearance under a given lighting condition. The contributions include refining the given but possibly coarse camera poses during optimization, representing the radiance field as a neural network, and a differentiable procedure of finding the ray-geometry intersections.

Strengths: Neural implicit representations of shape and appearance are promising new directions that overcome the high memory demand by more traditional explicit representations such as voxel grids. This paper demonstrates such use cases for both shape and appearance. Material-aware shape estimation also makes sense, because specularities, when modeled properly, should help inform the observer of the surface shape, rather than impede shape perception. This paper provides a concrete framework instantiating this general idea. The method compares with well-understood and strong traditional MVS baselines (the Colmap baseline in the paper).

Weaknesses: Overall, I am concerned on (1) the title and text overclaiming and being misleading, and (2) implicit modeling of geometry being an existing idea. Below I elaborate on these two points and some other weaknesses. (1) The title and many lines of text are stressing the (implicit) modeling of lighting and material (to the extent that I find delibrately misleading), but what the paper actually does is simply representing a radiance field with no decomposition of material and lighting at all. This was very confusing to me because I spent much time in trying to figure out whether this work factorizes lighting and materials from radiance field or not. I would strongly oppose mentioning lighting and material if no such factorization is done, and request explicit mentioning in the text that relighting cannot be done. (2) The main task of this paper is estimating geometry from multi-view (masked) images, by means of appearance modeling. In this sense, using an implicit representation of geometry is an important part of the contribution. But this is an existing idea starting from Occupancy Network "Occupancy Networks: Learning 3D Reconstruction in Function Space" and DeepSDF "DeepSDF: Learning Continuous Signed Distance Functions for Shape Representation". (3) A crucial advantage of this approach is its ability of optimizing cameras as the training progresses, but I don't see much improvement by doing so compared with the traditional MVS method. As Table 3 shows, the proposed method performs worse than COLMAP across different numbers of scans. Also, Figure 8 shows COLMAP gives smaller errors than IDR-optimized cameras, which seems to directly contradict the paper's claim. (4) In the video results, there seems to be undesired flickering (e.g., 0:38-0:40) in addition to the desired view-dependent effects. (5) DVR is said to be unable to handle complex lighting effects as in this work, but no qualitative comparisons were made to show that. This is a major missing piece. ==================== Post-rebuttal feedback: Thank you for addressing some of my concerns. I've raised my rating to 6 marginally above the borderline, conditioned on that the authors kept the promise to edit the title and text to clearly state lighting and materials are not factored and to remove the misleading claims. I hesitate to give a better rating than that because Fig. 8 shows the IDR-optimized cameras are worse than COLMAP cameras, and overall the camera optimization part is not well explored despite being a main contribution.

Correctness: The claims are overstated (please see weaknesses). The proposed method is valid and seems to have been implemented in a reasonable way empirically.

Clarity: There is much space to improve the clarity aspect of the paper. Many things remain unclear to me after reading the paper. For instance, what camera parameters are the authors opimizing for exactly? Given that the authors use "A global linear method for camera pose registration," I'd guess only camera extrinsics are being optimized for. The paper should make this clear to be standalone. Other confusions lie in Lemma 1, which shows the computuation of the intersection point between the ray and implicit geometry. Deferring the proof to the supplemental material is fine, but the authors should provide some intuition so that the reader can keep their flow going. The authors also need to make it clear that there's no factorization of materials and lighting, so no relighting is supported in "3.2 Approximation of the rendering equation" (see Weaknesses).

Relation to Prior Work: The reader of this paper would be interested to learn how this is similar to or different from the concurrent NeRF work, although this work won't have to be compared against NeRF in this cycle.

Reproducibility: No

Additional Feedback: The reader of this paper would be interested to learn how this is similar to or different from the concurrent NeRF work, although this work won't have to be compared against NeRF in this cycle.


Review 4

Summary and Contributions: This paper introduces Implicit Differentiable Renderer, a method able to learn 3D geometry, appearance and cameras from multiview RGB images. The core in the proposed method is an implicit scene representation and its differentiable rendering formulation. Experiments show that the proposed method can handle different objects with various materials.

Strengths: - The proposed method is able to reconstruct the geometry and the appearance simultaneously without relying on accurate camera calibration. The results are of good quality and impressive. - The proposed method is novel and based on sound theoretical groundings. The derivation are clear and complete.

Weaknesses: - The paper lacks discussion of and comparison with other neural rendering methods such as [A] and [B]. Although those methods do not learn the geometry explicitly, I think it is necessary to compare against them by evaluating the quality and accuracy of image rendering. - The authors only demonstrate the results of single objects. I wonder how well the method works on scenes with complex geometry. Is it possible to present the results on data of complex scenes such as the dataset in [C]? [A] DeepVoxels: Learning Persistent 3D Feature Embeddings. CVPR 2019 [B] Deferred Neural Rendering: Image Synthesis using Neural Textures. SIGGRAPH 2019 [C] Local Light Field Fusion: Practical View Synthesis with Prescriptive Sampling Guidelines. SIGGRAPH 2019 =========================================================== I appreciate that the authors addressed all reviews clearly in the response. Since the authors emphasized in the rebuttal that their method is aimed to reconstruct the geometry with masked RGB images, I won't regard the two weakness points mentioned above as "weaknesses" anymore. I read comments from other reviewers and the response, and I think the authors clarified their concerns . In summary, I am also positive on the paper and would like to keep my original rating.

Correctness: Yes

Clarity: Yes

Relation to Prior Work: Not enough. See Weaknesses.

Reproducibility: Yes

Additional Feedback: A comparison against NeRF[32] is recommended although I understand it is a concurrent work.

[Author Response · NeurIPS 2020]

We thank the reviewers for their insightful comments. We next address questions and comments raised in the reviews.

**R1, R3: There is no ability to disentangle lighting and material, the paper is misleading in that aspect.** We will
emphasize in the text that we only disentangle geometry and appearance, and that the appearance, which consists
of material (BRDF) and light, is not further factored. We will also add to the conclusions section a statement that
lighting and material are not separated, and mark it as an interesting future work. Therefore (and addressing the specific
request of R1), we suggest changing the paper title to: **Multiview Neural Surface Reconstruction via Disentangling**
**Geometry and Appearance**. In section 3.2 we will focus on approximating the surface radiance function $L(\boldsymbol{x}, \boldsymbol{n}, \boldsymbol{v})$
as a function of $\boldsymbol{x}$ (surface location), $\boldsymbol{n}$ (surface normal), and $\boldsymbol{v}$ (view direction). The rendering equation will only be
used to motivate the dependence of $L$ on $\boldsymbol{n}$.

**R1, R2: For fixed geometry, surface light fields are defined only in terms of location**
**($\boldsymbol{x}$) and view direction ($\boldsymbol{v}$) and are arbitrarily powerful, surface normals are therefore**
**not necessary.** In section 3.2 we will clearly state that, in theory, incorporating the surface
normal ($\boldsymbol{n}$) is not necessary for producing a general surface light field. However, it
is necessary for learning a general renderer ($M$) that is independent from *any specific*
*geometry*. We will incorporate an empirical evidence, such as the inset, that shows the effect
of incorporating normal in the renderer $M$. In this experiment we took two trained models
(consists of geometry network $f$ and renderer $M$), trained on two different DTU scenes,
once without and once with normals in the renderer. The inset shows: the reconstructed
geometry (left column); novel views using the trained renderer (middle column); and novel
views rendered using the renderer from the other scene (right column). Note that using the
normals in the renderer provides a better geometry-appearance separation: an improved
surface geometry approximation as well as correct rendering of different geometries.

*Without* normal

Geometry    Trained renderer    Other renderer

*With* normal (IDR)

Geometry    Trained renderer    Other renderer

**R2,R3: IDR cameras optimization vs. bundle-adjustment (e.g., Colmap SFM).** Our
method is a step towards end-to-end, simultaneous dense surface reconstruction and camera
optimization using 2D image supervision. This is in contrast to MVS pipelines (e.g.,
Colmap MVS) that cannot handle noisy cameras without a pre-processing step of bundle-
adjustment. In some cases, such as the "Fountain" scene, our method can go beyond
bundle-adjustment accuracy.

**R1, R2: Training and inference times are missing.** Timings were accidentally dropped. Training time are: 6.5 or 8
hours for 49 or 64 images, respectively, on a single Nvidia V100 GPU. Rendering (inference) time: 30 seconds for
$1200 \times 1600$ image with 100K pixel batches. All relevant details will be added to the text.

**R2, R3: Unexpected lighting effects in the supplied video.** In the original views
(training images), the camera body occasionally occludes some light sources, casting
temporary shadows on the object; see inset. Notice that in the video we move the
camera (i.e., viewing direction) and not the object, therefore when the camera moves
near such a view we see the projected shadow in the generated rendering.

**R1, R2: Clarify architecture details, supply data and code.** We will add exact architecture details to clarify how
equation 3 is implemented, and will make the code and data available as well.

**R2: "How were the 15 scans used in the results chosen?".** They were chosen arbitrarily to span a wide range of
different geometries and appearances. We did *not* cherry-pick results from a larger group of scans.

**R2: "It would be stronger result to show PSNR on a held-out set of images".** Indeed, in our experiments the
PSNR was computed over training images. Following the reviewer's question we performed an experiment where we
held-out 10% of the images as test views. Overall, we got 23.34 / 22.55 mean PSNR accuracy on the train / test images,
respectively. We will report the per-scene accuracies in the paper.

**R4: Comparison with other neural rendering methods.** In this paper we focus on reconstruction of geometry. The
mentioned papers focus on novel view generation. As the first inset above shows this is a different task. We therefore
chose to compare to methods with a similar objective; we will nevertheless add these references to our previous work
section.

**R4: Results on data of complex scenes.** It would be a very interesting future direction to handle non masked scenes.

**Additional answers for R3:** (1) DeepSDF and Occupancy papers do not learn geometry with 2D image supervision.
(2) We show qualitative comparisons with DVR in Figure 4, second column. (3) As commonly assumed in calibrated
SFM, we assume known intrinsics; we will clarify this in the text. (4) Intuition for Lemma 1 is given in lines 127-130.

[Meta-Review · NeurIPS 2020]

The paper leverages recent advances in neural scene representations to achieve solid 3D reconstruction results. While the method appears to be focused entirely on surface reconstruction and does not handle volumetric effects, the presentation and experimental evaluation are good. All reviewers recommend acceptance.